# Research on line loss analysis and intelligent diagnosis of abnormal causes in distribution networks: artificial intelligence based method



Yaohua Liao[1,2], Wang En[1,2], Bo Li[1,2], Mengmeng Zhu[1,2], Bo Li[1,2], Zhengxing Li[1,2] and ZhiMing Gu[1,2]

[1] Electric Power Research Institute, Yunnan Power Grid Co., Ltd., Kunming, Yunnan, China
[2] Yunnan Key Laboratory of Green Energy, Electric Power Measurement Digitalization, Control and Protection, Kunming, Yunnan, China

Corresponding author
Yaohua Liao,
liaoyaohua2023@163.com

## ABSTRACT

The primary source of energy losses in distribution networks (DNs) is rooted in line losses, which is crucial to conduct a thorough and reasonable examination of any unusual sources of line losses to guarantee the power supply in a timely and safe manner. In recent studies, identifying and analyzing abnormal line losses in DNs has been a widely and challenging research subject. This article investigates a key technology for the line loss analyses of DNs and intelligent diagnosis of abnormal causes by implementing artificial intelligence (AI), resulting in several prominent results. The proposed algorithm optimizes the parameters of the support vector machine (SVM) and suggests an intelligent diagnosis algorithm called the Improved Sparrow Search Algorithm and Support Vector Machine (ISSA-SVM). The ISSA-SVM algorithm is trained to calculate the data anomalies of line losses when changing loads and exhibiting exceptional performance to identify abnormal line losses. The accuracy of abnormality identification employing the ISSA-SVM algorithm reaches an impressive 98%, surpassing the performances of other available algorithms. Moreover, the practical performance of the proposed approach for analyzing large volumes of abnormal line loss data daily in DNs is also noteworthy. The ISSA-SVM accurately identifies the root causes of abnormal line losses and lowers the error in calculating abnormal line loss data. By combining different types of power operation data and creating a multidimensional feature traceability model, the study successfully determines the factors contributing to abnormal line losses. The relationship between transformers and voltage among various lines is determined by using the Pearson correlation, which provides valuable insights into the relationship between these variables and line losses. The algorithm's reliability and its potential to be applied to real-world scenarios bring an opportunity to improve the efficiency and safety of power supply systems. The ISSA that incorporates advanced techniques such as the Sobol sequence, golden sine algorithm, and Gaussian difference mutation appears to be a promising tool.

# INTRODUCTION

One of the most significant evaluation indicators for power supply firms is the magnitude of line losses, a visual representation of the power grid's scientific technology and operation status (*Wang, Liu & Ji, 2018*). The breadth and complexity of 10 kV DNs, which serve as a crucial link in the transmission of electric energy, have been growing, and so are line losses. According to data, the overall line losses of 10 kV DNs account for the highest share of the total line losses of the entire power grid system (*Huang et al., 2016*). The smaller the indicator's value is, the lower the power losses would be, which is beneficial to the robustness of the power grid and *vice versa*. This has a favorable effect on the performance of the grid system. However, because of the incomplete installation and data collection processes of low and medium-voltage stations and customer measurement meters, most conventional abnormality identification and evaluation of line losses are done by labor, which is very subjective. As a result, the identification process is not timely, has a high engineering volume, and has significant deviations in results (*Weizhou et al., 2019*).

The emergence of the big data age has given a wealth of data for investigating and computing DNs' line losses. Establishing a power grid data center allows for storing enormous amounts of historical data, allowing enterprise staff of power supply firms to conduct DNs' line loss research without being constrained by experience. Thus, the enormous amount of data could be utilized to perform several analyses, leading to laying the groundwork for a thorough analysis of DNs' line losses. Line loss analyses based on AI have steadily emerged as a contemporary research hotspot along with the steady advancements.

Current studies on the detection processes of abnormal line losses in DNs are limited, mainly based on data-driven methods. *Liu (2021)* suggested a 10 kV line abnormality diagnostic technique in light of the multiple influencing variables, the vast amount of data, and the complexity of diagnosis and analysis in the DNs' line losses. *Yao et al. (2021)* suggested a situation awareness-based diagnosis model for identifying and classifying different line loss abnormalities in the low-voltage DN (LVDN). *Li, Cai & Lou (2022)* developed an optimization solution for line losses in a 10 kV DN. An SVM and a hierarchical node identification strategy are used to address the problems of significant lack of information in the existing power-gathering process, severe mistakes in the concurrent line losses and computational procedures, and poor dependability in the course of line loss computations. By efficiently and promptly obtaining the injection current of different load nodes, the intelligent identification technique of load nodes may lower the line loss mistakes brought on by non-synchronous data. *Zhang (2019)* provided a technique for examining abnormal line losses and a strategy for handling them in a DN based on load characteristics. This approach employs hierarchical diversion and correlation analysis to determine the causes of abnormal line losses and their locations. *Hu et al. (2021)* provided an approach to computing real-time line losses based on equivalent DN resistance and established a set of equations for the resistance characteristics of low-voltage DNs. The issue is resolved by utilizing the total least squares approach, and the DN's equivalent resistance is discovered. The distribution transformer area's line losses

then vary in real-time with the user's electrical situation thanks to the clustering approach, which enables precise detection of impedance characteristics and aids power supply firms in spotting aberrant line loss conditions. After the time series data collected by the intelligent meters in the genuine electricity lines is verified, the distribution transformer region's daily instantaneous line loss curve is created. The suggested approach may be used to determine the real line loss of a low-voltage station based on sensor data and is independent of the DN's topological structure. *Zhou et al. (2018)* suggested a technique for detecting and correcting aberrant line loss data to increase the computation accuracy of the line loss rate. More up-to-date research and novel discussion can be found in *Zhang & Sun (2023)*, *Pop et al. (2022)*, *Lin, Tang & Fan (2022)*, *Houran et al. (2023)*, *Sicheng et al. (2022)*.

The approaches above suffer from issues including a long calculation time, a sizable computation volume, low accuracy rates, and low completion rates, unfavorable for quick completion of abnormal data detection tasks. In this study, a method called ISSA-SVM-based anomaly identification for large-scale line loss data daily in DNs is proposed, expecting that the suggested procedure may address the drawbacks of the available methods. By constructing a multidimensional feature anomaly traceability employing line change relationship, transfer of supply, and other power features, the causes of line loss data anomalies daily are detected by the ISSA-SVM to avoid long calculation time and volume problems. The ISSA-SVM is expected to address the shortcomings of the other available methods. Lastly, simulations are implemented to illustrate the viability of the suggested approach.

The motivation behind this study stems from the need to address the primary sources of energy losses in DNs, which are called line losses. Line losses not only lead to economic losses but also pose a significant challenge in ensuring timely, safe, and reliable power supply management. However, accurately identifying and analyzing the causes of abnormal line losses in DNs has proven to be a complex issue. The available methods often have difficulties in providing reliable and efficient solutions, highlighting the need to generate more innovative approaches.

Related issues:

1) The identification of data abnormality: One of the key challenges in line loss analyses is the accurate identification of data abnormalities. Conventional methods may fail to distinguish abnormal line loss patterns from the normal variations effectively. This study recognizes the need for a robust data preprocessing stage that normalizes line loss data daily and incorporates the notion of temporal dispersion to assess the level of irregularity. Thus, a more accurate and reliable identification of abnormalities in line loss data is reached.

2) AI and power systems: The application of AI in power systems has gained significant attention in recent years. This study leverages AI, specifically the ISSA and SVM to develop an intelligent diagnosis algorithm for abnormal line losses in DNs. Integrating AI algorithms allows for enhanced analysis, identification, and understanding of the complex relationships between various factors contributing to line losses.

3) Multidimensional feature traceability model: Identifying root causes behind abnormal line losses requires a comprehensive understanding of the factors influencing DNs. This study employs a multidimensional feature traceability model, combining different data types of power operations. By considering various aspects such as transformer conditions, voltage levels, and other relevant variables, a more holistic analysis can be conducted to identify the underlying causes of line losses.

4) Performance comparison and practical application: Comparing the performance of the proposed ISSA-SVM algorithm with others in the literature demonstrates its superiority and its practical effectiveness. The high accuracy rate of 98% achieved by the ISSA-SVM algorithm in identifying abnormal line losses highlights its potential for practical application in real-world scenarios. This is particularly significant given the increasing volume of line loss data in DNs, where more efficient analysis is essential for maintaining optimal system performance.

The study explores innovative approaches by incorporating AI techniques, developing an intelligent diagnosis algorithm, and utilizing a multidimensional feature traceability model. Improved abnormality identification, the application of advanced algorithms, and better practical performance contribute to advancing line loss analyses in power DNs. Therefore, systems' efficiency and reliability could be enhanced.

The structure of the article is outlined as follows: "A model based on multidimensional feature anomaly traceability for daily distribution network line loss data" discusses the model used for anomaly traceability of distribution networks using multidimensional features. The proposed method is presented in "The issa-svm-based anomaly identification method for large-scale line loss data daily in DNS". "Experimental analysis" is allocated to experimental studies and research findings. Results and discussion are presented in "Results and discussion". "Conclusion" concludes the conducted research.

## A MODEL BASED ON MULTIDIMENSIONAL FEATURE ANOMALY TRACEABILITY FOR DAILY DISTRIBUTION NETWORK LINE LOSS DATA

Conventional abnormality identification of DNs based on daily line loss data is primarily conducted manually, with a lot of screened data, no involved advanced technologies, and almost little utility for abnormality identification. Depending on the thorough data mining techniques, atypical line-transformer interactions, and issues with multiple power supplies are two potential sources of data anomalies for daily line losses.

The data is not updated promptly throughout the construction, operation, and maintenance of the line, disrupting the relative connection between the line and transformer and producing aberrant line loss data. A dual power supply denotes that a high-voltage user has two or more metering points, each corresponding to a line in turn. However, due to errors in the system file records, real operation, and other factors, such as the line transfer for a reverse operation, which causes a line outage that requires maintenance, staff will switch the distribution routes to guarantee the credibility of the

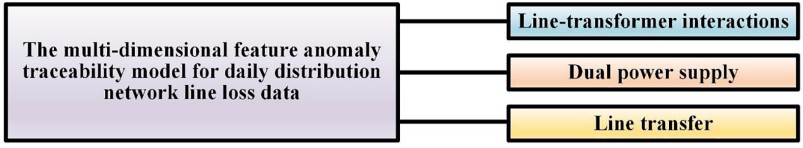

**Figure 1 The multidimensional feature anomaly traceability model for daily distribution network line loss data.**                               

power supply. The switchover, which alters file information and impacts how the DN calculates statistics of daily line losses, is often a brief procedure.

Figure 1 depicts a traceability model based on a multidimensional feature to detect daily anomalies for line loss data of DNs daily.

The multidimensional feature anomaly traceability model for daily DN's line loss data is designed to identify and trace anomalies in line loss data in a DN. Line losses refer to the difference between the energy supplied into a DN and the energy consumed by end users. Anomalies in line loss data could indicate equipment malfunctions, theft, or metering errors.

The model analyzes multiple dimensions or features of the line loss data, such as time of day, geographical location, load profile, weather conditions, and other relevant parameters. The model can capture complex relationships and patterns that may indicate anomalies by considering multiple dimensions.

A high-level overview of the steps involved in the construction of the model is presented as follows:

1) Data collection: Gather daily line loss data from the DN. This data should include information on various dimensions or features relevant to line losses.

2) Data preprocessing: Cleanse the data by removing any outliers, missing values, or errors. Normalize the data if necessary to ensure all features are on a similar scale.

3) Feature engineering: Extract relevant features from the data or construct new features that could potentially improve anomaly detection. For example, each feature's statistical measures could be computed such as mean, standard deviation, or skewness.

4) Model training: Utilize anomaly detection algorithms, such as clustering, statistical methods, or machine learning techniques, to train a model on the preprocessed data. The model is expected to learn regular patterns in the line loss data and identify deviations from those patterns as anomalies.

5) Model evaluation: Assess the performance of the trained model employing evaluation metrics such as precision, recall, F1-score, or area under the receiver operating characteristic curve (AUC-ROC). Adjust the model parameters or run experiments with different algorithms to improve performance.

6) Traceability and interpretation: Once an anomaly is detected, the model should provide traceability, *i.e.*, the ability to track back and identify the specific dimension or feature that contributed to the problem. This traceability enables operators to investigate the root causes of the anomaly and take appropriate corrective actions.

7) Integration and deployment: Integrate the trained model into the daily operations of DNs. By doing so, incoming line loss data in real-time is processed, any anomalies are flagged and traceability information is provided to operators or automated systems.

By examining the collection system of the power information, extracting the feature vectors of line loss data, and studying the correlation of all transformer voltages between various lines using the Pearson correlation, the grid's line loss of loading data daily, such as voltage, current, and power, is obtained.

$$C_{TV} = \begin{bmatrix} C_{TV,11} & \dots & C_{TV,1n} & C_{TV,1l} \\ \vdots & \vdots & \vdots & \vdots \\ C_{TV,n1} & \dots & C_{TV,nn} & C_{TV,nl} \end{bmatrix} \tag{1}$$

where $C_{TV}$ represents the voltage correlation characteristic, $C_{TV,ij}$ denotes the voltage correlation between transformer $i$ and transformer $j$ under the line, and $C_{TV,il}$ represents the voltage correlation between transformer $i$ and line $l$.

The voltage correlation matrix cannot be easily included in the multidimensional feature model due to the differences in the number of distribution substations in each line. To improve the precision of the analysis of abnormal data traceability, it is necessary to implement the column aggregation operation on the voltage correlation features obtained from the operation and compute the highest (HV), lowest (LV), mean (MV), maximum (MAXV), minimum (MINV), skewness (SV), kurtosis values (KV), respectively and standard deviation (SD) within each line statistics to form the voltage correlation matrix based on statistics. Equation (2) presents the matrix.

$$M = \begin{bmatrix} M_{i,mean} & M_{i,std} & M_{i,max} & M_{i,min} & M_{i,skew} & M_{i,kurt} \\ \vdots & \vdots & \vdots & \vdots & \vdots & \vdots \\ M_{n,mean} & M_{n,std} & M_{n,max} & M_{n,min} & M_{n,skew} & M_{n,kurt} \end{bmatrix} \tag{2}$$

where $M_{i,mean}, M_{i,std}, M_{i,max}, M_{i,min}, M_{i,skew}, M_{i,kurt}$ represent the MV, SD, MAXV, MINV, SV, and KV of the Pearson correlation index for transformer, $I$ and the same line, showing the voltage correlation distribution characteristics of transformer and line. Equation (3) presents the calculation of $M_{i,mean}$.

$$M_{i,mean} = \frac{\sum\limits_{n}^{i=1} C_{TV,ij} + C_{TV,il}}{n+1} \tag{3}$$

At the same time, extrapolating the voltage data statistics of the transformer itself values:

$$V = \begin{bmatrix} V_{i,mean} & V_{i,std} & V_{i,max} & V_{i,min} & V_{i,skew} & V_{i,kurt} \\ \vdots & \vdots & \vdots & \vdots & \vdots & \vdots \\ V_{n,mean} & V_{n,std} & V_{n,max} & V_{n,min} & V_{n,skew} & V_{n,kurt} \end{bmatrix} \tag{4}$$

where $V_{i,mean}, V_{i,std}, V_{i,max}, V_{i,min}, V_{i,skew}, V_{i,kurt}$ represents the MV, SD, MAXV, MINV, SV, and KV of the voltage ratio of transformer $i$ to its line $l$ in the specified period. $V_{i,mean}$ is computed in Eq. (5).

$$V_{i,mean} = \frac{1}{m} \sum_{n}^{k=1} \frac{U_{i,k}}{U_{l,k}} \qquad (5)$$

where $U_{i,k}$ can be defined as the voltage value of the transformer at time $k$; $U_{l,k}$ can be defined as the voltage value of line $l$ at time $k$.

Thus, DN's daily line loss details can be deduced, and the source of abnormal data of line loss can be obtained to facilitate the improvement of supply line defects.

# THE ISSA-SVM-BASED ANOMALY IDENTIFICATION METHOD FOR LARGE-SCALE LINE LOSS DATA DAILY IN DNS

## Support vector machine (SVM)

SVM is a widely researched and implemented method to run classifications. Its core idea is to map from the input space to the high-dimensional space to find an appropriate optimal classification hyperplane to split samples. The derived optimization objective function of the SVM is shown in Eq. (6), and its constraint function is denoted by $y_i(w^T \Phi(x_i) + b) \geq 1$. The target of identifying the maximum value is converted into a target of identifying the extreme value, and the Lagrange multiplier is used to resolve it (*Hosseini, Mahoor & Khodaei, 2018*; *Obiedat, 2022*; *Qu et al., 2019*).

$$\arg \max_{w,b} \frac{1}{||\mathbf{w}||} \qquad (6)$$

There are a few samples of anomalous distributions in reality. Once the relaxation variable $\xi$ is added, the spacing issue is known as soft spacing. This is because it allows for extreme samples by assigning a matching relaxation variable. Equation (7) illustrates the derivation after the relaxation factor is added to the constraint in Eq. (6), resulting in a tight categorization. It cannot contain mistakes if $c$ tends to be quite big. Larger mistakes can be accepted when $c$ tends to be extremely tiny, and $c$ is a parameter that must be chosen based on the particular case (*Zhang et al., 2018*).

$$\min \frac{1}{2} ||w||^2 + c \sum_{n}^{i=1} \xi_i \qquad (7)$$

$\Phi(x)$ represents a transformation method, *i.e.*, kernel function transformation. When the data set cannot be divided linearly, the kernel function must connect the data set to a space with high dimensions where it can be divided linearly. The most often used kernel function is the Gaussian RBF, defined in Eq. (8), where that of the parameter controls the regional radial action range of the Gaussian RBF. The sample distribution is more dispersed when the value of $\sigma$ is large. When $\sigma$ tends to be very small, the sample density is higher. When $\sigma$ tends to be minor, the sample density is more concentrated, which leads to overfitting problems. Therefore, the primary goal of the SVM model is to choose the correct kernel function parameters.

It was challenging to accomplish optimal SVM computation in the past since the penalty factor and kernel function parameters were frequently specified manually. This research offers the ISSA to achieve the optimization for parameter search of the SVM.

## Improved sparrow search algorithm (ISSA)

The SSA is a brand-new group intelligence algorithm inspired by the foraging, anti-predation, and vigilance behaviors of sparrows, which mainly consists of a discoverer, an entrant, and a scout. The SSA algorithm consists of a discoverer, a joiner, and a scout. The discoverer is in charge of foraging, and the joiner will follow the best-adapted discoverer to obtain food, and the joiner will watch the discoverer to grab food. Once a scout senses a predator in danger during foraging, it will quickly alert the entire population to fly to a secured location (*Sun, 2022*). The discoverer location can be updated as shown in Eq. (8).

$$X_{i,j}^{t+1} = \begin{cases} X_{i,j}^t \cdot \exp\left(\dfrac{-i}{a \cdot iter_{\max}}\right), & R_2 < ST \\ X_{i,j}^t + Q \cdot L & , \quad R_2 \cdots ST \end{cases} \tag{8}$$

where $iter_{\max}$ denotes the maximum number of iterations, $X_{i,j}^{t+1}$ denotes the position of the $i$-th sparrow in the $t$-th iteration, and the position of the $i$-th sparrow in the $j$-th dimension, $a$ shows a random number in $(0,1]$. $Q$ denotes a random variable following the standard normal distribution, $L$ denotes a $1 \times d$ matrix consisting of 1s, and $d$ represents the dimensionality of the objective function. $R_2 \in [0,1]$, $ST \in [0.5,1]$, and denote the alarm value and the safety threshold, respectively (*Xu et al., 2022*).

The joiner position is updated as shown in Eq. (9)

$$X_{i,j}^{t+1} = \begin{cases} Q \cdot \exp\left(\dfrac{X_{worst}^t - X_{i,j}^t}{a \cdot iter_{\max}}\right) & , \quad i > n/2 \\ X_p^{t+1} + \left|X_{i,j}^t - X_p^{t+1}\right| \cdot A^+ \cdot L , & \text{otherwise} \end{cases} \tag{9}$$

where $X_{i,j}^{t+1}$ denotes the best position of the discoverer in the $(t + 1)$ generation, $X_{worst}^t$ denotes the worst position in the population of the $t$-th generation, A denotes the $1 \times d$ matrix composed of only 1 or −1, and $A^+ = A^T(AA^T)^{-1}$.

The scout position is updated as shown in Eq. (10)

$$X_{i,j}^{t+1} = \begin{cases} X_{best}^t + \beta \cdot \left|X_{i,j}^t - X_{best}^t\right|, f_i > f_g \\ X_{i,j}^t + K \cdot \dfrac{\left|X_{i,j}^t - X_{worst}^t\right|}{(f_i - f_w) + \varepsilon}, f_i = f_g \end{cases} \quad \text{mc} \tag{10}$$

where $X_{best}^t$ represents the optimal position in the $t$-th generation of the population, $\beta$ denotes a random number with a normal distribution with mean 0 and variance 1, $K \in [-1,1]$ is a random number, $f_i$ represents the fitness value of the $t$-th sparrow, $f_g$ denotes the optimal fitness value of the current sparrow population, and $f_w$ represents the worst fitness value of the current sparrow population.

The challenge for the SSA is to balance local and global searches and avoid being caught in a local optimum. The optimization strategy is implemented as follows:

(1) Sobol sequence initialization

Sobol sequence is a low-disparity sequence that focuses on producing uniform distribution in the probability space compared with a random number sequence. In this article, we use the Sobol sequence to initialize the original location of the population, which effectively improves the diversity of the initial solution and helps improve the algorithm's convergence speed and the ability of global search.

When initializing the population's original location employing the Sobol sequence, the algorithm benefits from several advantages:

1) Improved diversity of initial solutions: The Sobol sequence provides a more even distribution of points compared to a random number sequence. This helps to ensure that the initial population covers a broader range of the search space, increasing the diversity of the solutions. Having diverse initial solutions can be advantageous as it allows the algorithm to explore different regions of the search space, potentially leading to a better exploration of the solution landscape.

2) Accelerated convergence speed: The Sobol sequence initialization can help the algorithm converge faster toward an optimal solution by providing a more evenly distributed set of initial solutions. This is because the population starts with solutions that cover a more significant portion of the search space, allowing the algorithm to quickly identify promising regions and focus on its search efforts more efficiently.

3) Enhanced global search ability: The Sobol sequence initialization aids in improving the algorithm's global search capabilities. By spreading the initial solutions uniformly across the search space, the algorithm is less likely to miss essential regions or get trapped in local optima. This enables the algorithm to explore a broader range of potential solutions and increases the chances of finding the global optimum or better approximations.

Overall, by utilizing the Sobol sequence to initialize the population's original location, the algorithm benefits from improved solution diversity, accelerated convergence speed, and enhanced global search ability. These advantages contribute to the effectiveness of the optimization process and increase the likelihood of finding high-quality solutions more efficiently.

(2) Golden sine algorithm (Golden-SA)

The Gold-SA, which has the benefits of a straightforward theory and robust search capabilities, shrinks the space for searching by the golden ratio to approach the method's ideal answer.

The position update equation in Gold-SA is presented as follows.

$$X_{i,j}^{t+1} = X_{i,j}^t \cdot |\sin(r_1)| + r_2 \cdot \sin(r_1) \cdot \left| k_1 \cdot X_p^t - k_2 \cdot X_{i,j}^t \right| \tag{11}$$

where $X_p^t$ represents the optimal position of the individual in the $t$-th generation.

To enhance the global search capabilities in the initial phase and the local development capabilities in the later stage of the optimization process, the discoverer in the SSA requires a wider search span initially, followed by a smaller search range to mine the global optimal place.

In the conventional SSA, as the number of iterations increases, the search range of each dimension of the sparrow population shrinks, potentially limiting the algorithm's ability to perform effective global search. To address this limitation, a modification called Gold-SA is proposed to balance the global search capabilities in the initial phase with the local development capabilities later.

In the Gold-SA, Eq. (8) of the original SSA is improved to achieve this balance. By adjusting the parameters and updating the equation, the Gold-SA ensures that the initial phase of the discoverer's search field retains a broader range, enabling it to conduct global searches effectively. On the other hand, the later-stage capabilities of regional growth are also maintained to facilitate local development.

By matching the initial phase capability for global search with the latter stage capability for regional growth, the Gold-SA aims to strike a balance between exploration and exploitation throughout the optimization process. This approach allows the algorithm to explore a broader search space in the beginning, enabling it to discover promising regions and focus on the search in later stages to refine solutions and converge towards the global optimum or better approximations.

The modification introduced in the Gold-SA aims to improve the overall search capabilities of the algorithm by addressing the limitations of the original SSA and ensuring that the algorithm is effective in both global and local search tasks at different stages of the optimization process.

After introducing the Gold-SA, the discoverer's position is updated as follows

$$X_{i,j}^{t+1} = \begin{cases} X_{i,j}^t \cdot |\sin(r_1)| + r_2 \cdot \sin(r_1) \cdot \left|k_1 \cdot X_p^t - k_2 \cdot X_{i,j}^t\right| & , R_2 < ST \\ X_{i,j}^t + Q \cdot L & , R_2 \cdots ST \end{cases} \tag{12}$$

(3) Gaussian difference variation

In the iterative process, to prevent the population from falling into the local optimum problem, the optimal position of the population is perturbed at each iteration employing Gaussian difference variation. When compared with the conventional difference variation algorithm, Gaussian difference can generate a more significant perturbation in the vicinity of the current individual variation, which makes it the algorithm to jump out of the local optimum and employ the greedy strategy to update the optimal position of the population if the position is better after the perturbation; otherwise, it remains unchanged. The Gaussian difference variation equation is presented as follows:

$$L^t = k_1 \cdot g_1 \cdot \left(X^* - X_{best}^t\right) + k_2 \cdot g_2 \cdot \left(X_{rand} - X_{best}^t\right) \tag{13}$$

where $k_1$ and $k_2$ denotes weight coefficients, $g_1$ and $g_2$ represent Gaussian distributed random variables with 0 means and 1 variances, $X^*$ denotes the population optimal

sparrow position, $X_{rand}$ denotes the random sparrow position, and $L^t$ denotes the position after Gaussian difference variation.

After obtaining the location of the perturbation, the adaptation values of the locations before and after the perturbation are compared. The position with optimal fitness is selected as the optimal position for this iteration of the population. The optimal position can be updated as shown in Eq. (14).

$$X_{best}^t = \begin{cases} L^t, f(L^t) < f(X_{best}^t) \\ X_{best}^t, f(L^t) \geq f(X_{best}^t) \end{cases} \tag{14}$$

$f(L^t)$ and $f(X_{best}^t)$ denote the adaptation values of the post-disturbance and pre-disturbance positions, respectively.

The iterative process described aims to prevent the solution from getting stuck in a local optimum during optimization. To achieve this, the optimal position of the population is perturbed by using a technique called Gaussian difference variation.

In the conventional difference variation algorithms, perturbations are typically generated by using a fixed range or a uniform distribution. However, Gaussian difference variation is employed, which introduces a larger concern near the current best solution.

Using a Gaussian distribution, the perturbations are more likely to occur closer to the current optimal solution. This allows the algorithm to explore a larger area around the current solution, increasing the chances of finding a better solution and escaping local optima.

After perturbing the current best solution, the algorithm evaluates the fitness or quality of the perturbed solution. Suppose that the perturbed solution provides an improvement (*i.e.*, a better fitness value) compared to the previous best solution. In that case, the optimal position of the population is updated to the perturbed solution by utilizing a greedy strategy.

If the perturbation leads to an improved solution, the algorithm greedily accepts the new solution as the new optimal position. However, if the perturbed solution does not improve the fitness value, the optimal position remains unchanged, ensuring the population does not regress to a worse solution.

By incorporating Gaussian difference variation and a greedy update strategy, the algorithm aims to balance both exploration and exploitation and explores the search space by perturbing solutions in the vicinity of the current best solution, and exploits the improvements by greedily updating the population's optimal position when a better solution is found.

This approach helps to enhance the algorithm's ability to escape local optima and potentially find a global optimal solution or a better approximation in the optimization process.

## The detection process of abnormal data of line losses

In this article, the search area of the SVM parameter $\sigma$ is set to [0.001, 1] and that of parameter c is set to [1, 100]. The flow of searching the optimal parameter combination of

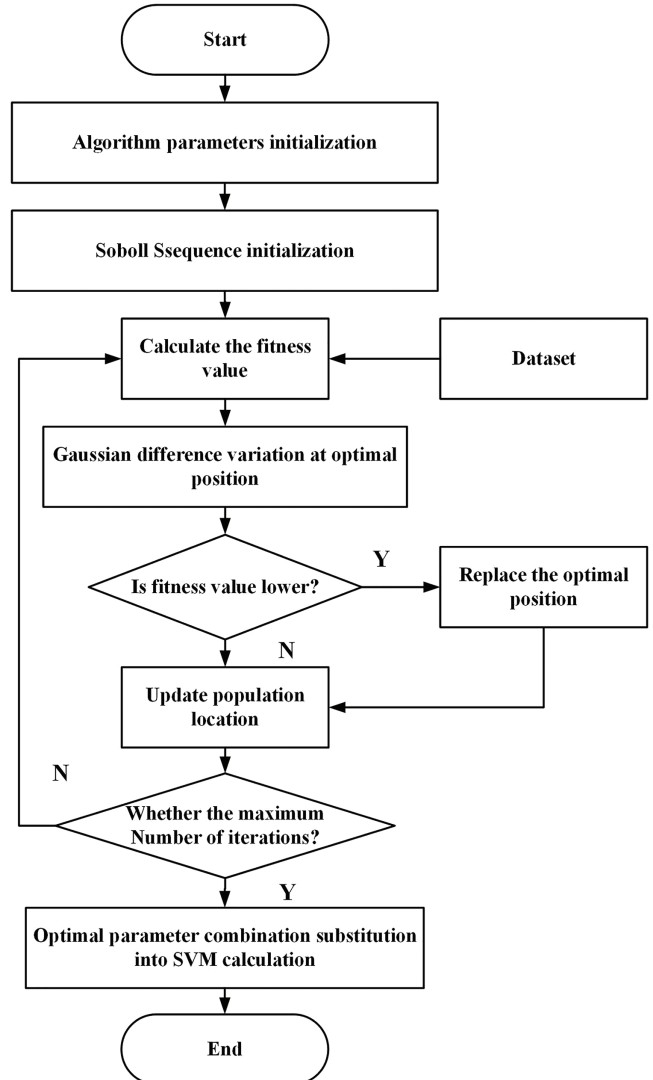

**Figure 2** **The flow of searching the optimal parameter combination of SVM using ISSA for line loss abnormal data detection.**

the SVM using the ISSA for the abnormal data detection of line losses is shown in Fig. 2, and the specific steps are given as follows:

The proposed algorithm

Step 1: Initialize the parameters of the ISSA, including the number of species, the number of discoverers and detectors, the search dimension, the search range, and the maximum number of iterations.

Step 2: Initialize the location of the sparrow population using the Sobol sequence and employ the anomaly detection error rate as the fitness value of the proposed algorithm.

Step 3: Perturb the optimal result according to Eq. (14), compare the fitness values before and after the perturbation, and select the one with the better fitness value as the optimal result.

Step 4: According to Eqs. (9), (10), and (12), update the positions of discoverer, joiner, and scout, respectively.

Step 5: If the maximum number of iterations is not reached, return to Step 3 and continue the search. If the maximum number of iterations is reached, the search ends, and the optimal combination of parameters is output.

Step 6: The trained ISSA-SVM model is employed to diagnose the line loss data.

The combination of the ISSA and the SVM aims to improve the effectiveness of detecting abnormal data for line losses in a DN. The SVM is a widely used machine learning algorithm for classification tasks, known for its ability to handle complex patterns and high-dimensional data.

In this context, the SVM parameters $\sigma$ and c play a crucial role in determining the performance of the SVM model. The parameter $\sigma$ represents the width of the Gaussian kernel used in the SVM, which controls the smoothness of the decision boundary. The parameter c represents the regularization parameter that balances the trade-off between achieving a low training error and a low complexity model.

The ISSA is employed to search for the optimal combination of $\sigma$ and c concurrently, maximizing the detection accuracy for abnormal data of line losses. The ISSA explores the parameter space by initializing a population of potential solutions (sparrows) and iteratively updating their positions. It evaluates the fitness of each solution based on the anomaly detection error rate.

Through the iterative search process, the ISSA perturbs the current optimal solution, compares the fitness values, and updates the positions of discoverers, joiners, and scouts. This allows the proposed algorithm to dynamically adapt its search strategy and explore different regions of the parameter space to find a better combination of $\sigma$ and c concurrently.

The objective is to identify the optimal combination of $\sigma$ and c that minimizes the error rate in detecting abnormal data of line losses. Once the optimal parameter combination is found, the trained ISSA-SVM model can be implemented to classify and diagnose line loss data in DNs, enabling the identification of anomalies such as equipment malfunctions, theft, or metering errors.

By leveraging the ISSA's optimization capabilities and SVM's classification abilities, the proposed method aims to enhance the accuracy and effectiveness of abnormal data detection of line losses, ultimately contributing to improved operational efficiency and maintenance in a DN.

# EXPERIMENTAL ANALYSIS

## Data source

Twenty DN segments in region A are the research's target area, and 14 station areas serving 7,854 consumers are utilized as the experimental sample data to run the simulation. The zones are designated 1 through 14, with zones 1–6 being urban residential zones, 7–8 being commercial, 9 through 10 being industrial, and 11 through 14 being agricultural. The topology structure is shown in Fig. 3. The DN comprises 10 10 kV low voltage stations and one 10 kV outlet line with 20 line nodes. A zone's daily line loss value

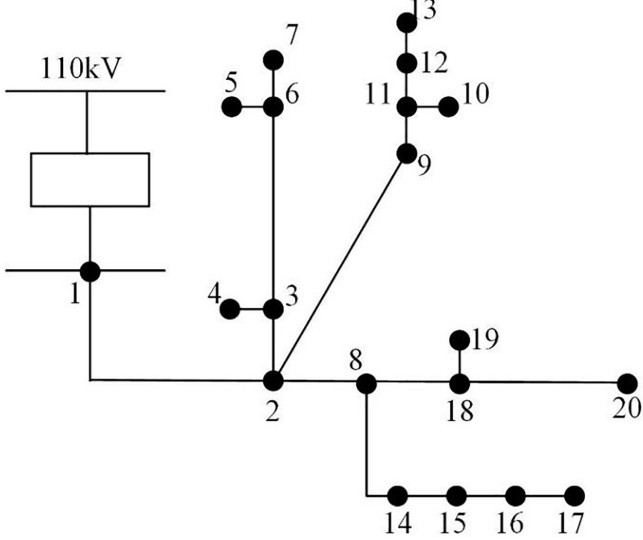

**Figure 3** DN topology.  

is calculated by using the difference between the electricity of various zones and the total electricity used by users inside the zone. Each zone border chamber is equipped with a zone energy meter. Table 1 displays some of the data from the station area loss statistics. Among them, 1,050 sets of line loss data in seven station areas from 2017 to 2022 are utilized as samples. The loss causes of each line correspond to 150 samples. The training and test sets are split in a 4:1 ratio, yielding 840 sets of samples for the training set and 210 sets for the test set, respectively.

Following the actual local loss status, abnormal traceability markers are set, as shown in Table 2.

## The analysis of the algorithm accuracy

The ISSA-SVM is compared with the identification results of line loss causes of BP neural network (BPNN) (*Tang et al., 2018*), Extreme Learning Machine (ELM) (*Yang, Zhao & Jing, 2020*), SVM, PSO-SVM (*Tang et al., 2018*), and the SSA-SVM, respectively, and the upper bound of iterations is set to 100. The obtained identification accuracy is shown in Fig. 4.

Figure 4 depicts that when compared with BPNN and ELM, the SVM has a slightly more substantial learning effect for samples, which makes the final recognition accuracy higher than BPNN and ELM. The SVM can dig sample features by in-depth learning, so the recognition accuracy of the SVM is 87.8912%, which is higher than the diagnosis recognition accuracy of 84.3537% of the BPNN and 86.0544 of the ELM, respectively. The SVM optimized by the SSA can effectively overcome the deficiency of the SVM that relies on the selection of parameters manually, which can maximize the classification accuracy of the SVM, making the final recognition accuracy as high as 91.6327%. The comparison between the SSA-SVM and the PSO-SVM shows that the SSA-SVM can effectively avoid falling into the local optimum, so the recognition accuracy is better. After the improvement of the SSA, the recognition accuracy of the ISSA-SVM reaches 98.9796%,

**Table 1 Some of the data from the station area loss statistics table.**

| Station number | Electricity supply/kW-h | Monthly line loss | Referenceable line loss rate/% |
|---|---|---|---|
| 1 | 2,358.20 | 8.22 | 2.64 |
| 2 | 1,354.60 | 2.7 | 2.36 |
| 3 | 4,230.00 | 1.89 | 1.89 |
| 4 | 1,866.00 | 2.34 | 2.42 |
| 5 | 742.50 | 2.32 | 0.01 |
| 6 | 1,916.66 | 2.92 | 2.42 |
| 7 | 1,423.20 | 2.51 | 2.55 |
| 8 | 896.10 | 2.27 | 2.47 |
| 9 | 822.50 | 2.19 | 1.72 |
| 10 | 1,697.30 | 1.18 | 1.98 |
| 11 | 1,148.70 | 2.79 | 1.99 |
| 12 | 2,120.30 | 2.23 | 2.23 |
| 13 | 1,074.20 | 2.14 | 1.89 |
| 14 | 4,108.00 | 2.65 | 1.78 |

**Table 2 Causes of abnormal daily line loss data of power grid.**

| Tag code | Reason for abnormality | Daily line loss status |
|---|---|---|
| 1 | Line transformer relationship abnormal | Low negative loss |
| 2 | Dual power supply problem | High negative loss |
| 3 | Line transfer | Micro-high negative loss |
| 4 | Theft of electricity | High negative loss |
| | | Very high negative loss |
| 5 | Abnormbd268mca39l meter base | Micro-high negative loss |
| | | High negative loss |
| | | Very high negative loss |
| 6 | Energy meter deviation | Low negative loss |
| | | Normal |
| 7 | Aging of line equipment | Normal but high |

which shows that the improvement of the SSA is capable of regulating the local search and global search capabilities concurrently and effectively, thus improving the classification accuracy of the proposed algorithm, which has significant advantages when compared with other algorithms and is suitable for the detection and recognition of abnormal data of line losses in DNs.

(2) Operation efficiency

In the anomaly data detection of line losses in DNs, the operational efficiency of the proposed model is also a critical assessment index. To confirm even further the operational efficiency of the ISSA-SVM, the article compares the operational efficiency of the ISSA-SVM with those of the BPNN, ELM, SVM, PSO-SVM, and SSA-SVM, respectively. It

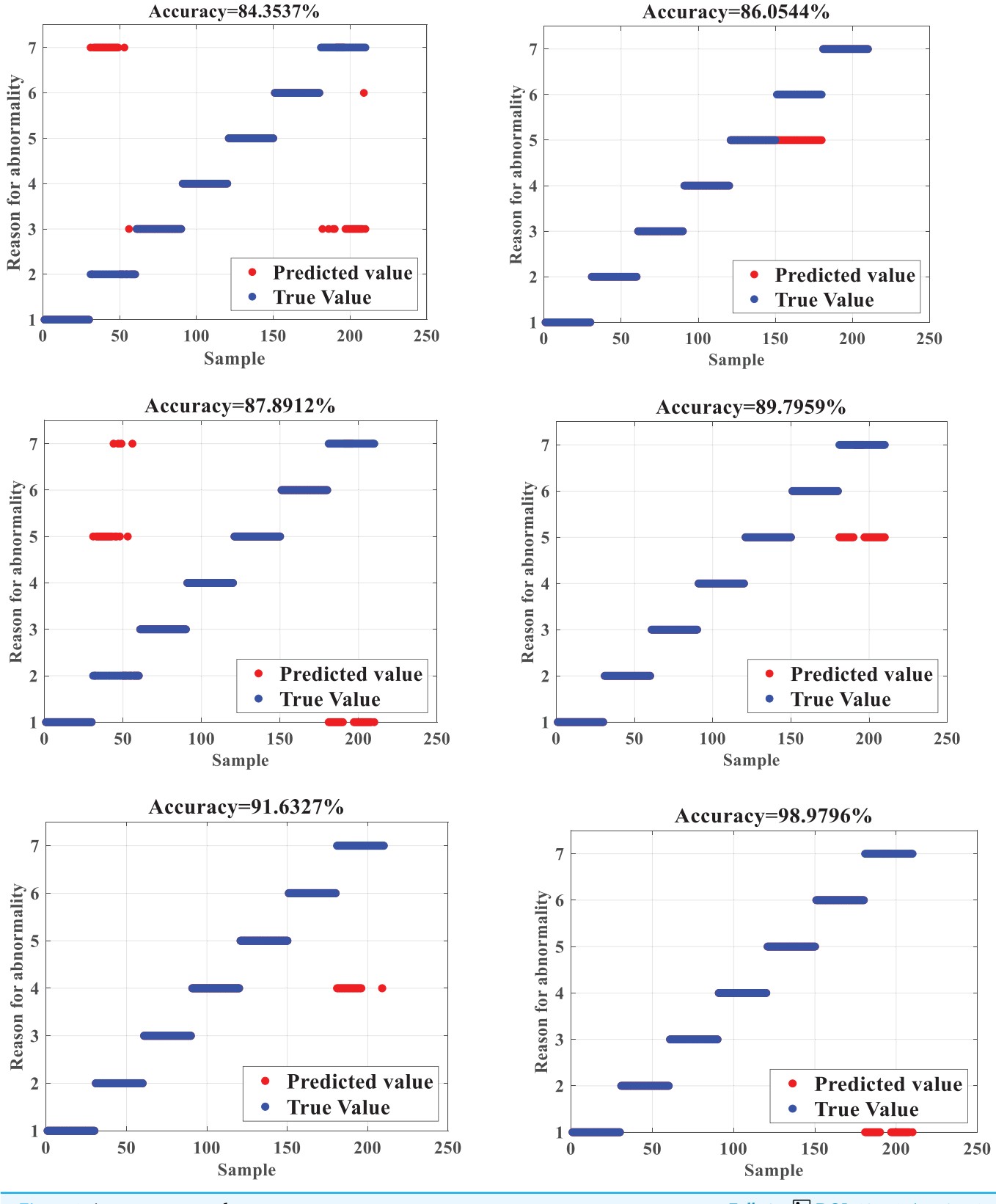

**Figure 4  Accuracy test results.**

**Table 3 Operation time test results.**

| Algorithm | Operational time |
|---|---|
| BPNN | 36.4s |
| ELM | 28.2s |
| SVM | 26.8s |
| PSO-SVM | 42.6s |
| SSA-SVM | 35.8s |
| ISSA-SVM | 27.6s |

focuses on the operational efficiency of the proposed algorithm from the perspective of operational time when performing classification operations on the test set, and the obtained results are shown in Table 3.

Table 3 depicts that while dealing with high-dimensional data derived from DNs with a small number of line losses, the operation time of the BPNN is long due to the limitation period of its network, and the operation times of both SVM and ELM are relatively shorter. While both SSA and PSO are a concern, after optimizing the SVM, the computing time also increases due to the increase of the computing process when compared with that before the SVM optimization (*Wu et al., 2019*; *Wu et al., 2021*). However, the ISSA improves the global optimization efficiency of the SVM so that the ISSA-SVM does not increase the operation time too much while ensuring high recognition accuracy and achieves a reasonable balance between accuracy and efficiency, which has significant advantages when compared with other algorithms.

## RESULTS AND DISCUSSION

The effectiveness of an AI system, including the analysis and diagnosis of line loss anomalies of DNs, depends significantly on data availability and quality. The accuracy and reliability of AI models' outputs are directly influenced by the data implemented for the training stage of the model.

In the context of the analyses of line losses in DNs, accurate and comprehensive data availability is crucial. The data should cover various scenarios, encompassing various types of line losses and their associated causes. Additionally, the data should be up-to-date to reflect the current operating conditions of the DN.

If the data utilized for training the AI model is incomplete, outdated, or contains errors, it can adversely affect the analysis and diagnosis processes. A few potential challenges are summarized as follows:

1) Limited representation of anomalies: Incomplete or insufficient data may not adequately represent the diverse range of abnormalities contained in DNs, which could result in the limitations of AI models to identify and diagnose specific line loss anomalies.

2) Bias in the training data: If the training data is biased towards certain types of anomalies or specific network conditions, the AI model may exhibit similar biases in its analysis

and diagnosis, which could lead to inaccurate or incomplete insights and recommendations.

3) Reduced generalizability: Outdated or non-representative data may limit the generalizability of the findings generated by AI models. The model may not perform well when applied to new or unseen data, hindering its effectiveness in real-world scenarios.

To mitigate these challenges, it is important to ensure the availability of high-quality data. This can involve regular data collection and maintenance processes, data validation techniques, and robust quality assurance practices. Additionally, efforts should be made to gather data from diverse sources and locations to capture a comprehensive view of the DN's operation and line loss phenomena.

By addressing data limitations and striving for accurate and comprehensive data, analyzing and diagnosing line loss anomalies of DNs can be improved, leading to more reliable and valuable insights for effective network management.

DNs can indeed vary significantly in terms of their scale, infrastructure, geographical location, and operational characteristics. As a result, the findings from a specific research study may not be directly applicable to all DNs.

Different DNs' unique characteristics and complexities can impact the behavior of line losses, so the underlying causes of anomalies, and the effectiveness of anomaly detection and diagnosis techniques could change. Factors such as network topology, load patterns, types of equipment, weather conditions, and regulatory frameworks can all play a role in shaping the DN's behavior.

Therefore, it is essential to recognize the need for further validation and adaptation of research findings to different contexts of DNs. This could involve conducting additional studies or experiments tailored to the target network system, considering its specific characteristics and operational requirements.

Furthermore, collaboration and knowledge-sharing among researchers, industry practitioners, and domain experts can facilitate a deeper understanding of line losses for a DN and the development of context-specific solutions. This collaborative approach can help address the challenges and complexities of generalizing research findings to diverse DN systems.

By conducting further validation and customization, researchers and practitioners can ensure that the insights and techniques derived from studies are effectively applied and adapted to the specific requirements and characteristics of different DNs.

Two important limitations related to the interpretability of AI models and the dynamic nature of DNs.

1) Lack of interpretability: AI models, particularly complex ones like deep learning algorithms, often lack interpretability. While they can provide accurate predictions or diagnoses, understanding the underlying reasons or factors contributing to those results can be challenging. This limitation can make understanding the root causes of abnormal line losses in DNs difficult. Interpretability is crucial for building trust in the AI system

and effectively addressing and resolving the identified issues. Researchers and practitioners should explore approaches such as explainable AI techniques or the features important analysis to enhance the interpretability of the models and provide meaningful insights into the causes of abnormal line losses.

2) Dynamic and evolving nature of DNs: DNs are continuously evolving highly active systems. Changes in load patterns, infrastructure upgrades, equipment replacements, and other factors can significantly impact line losses and the underlying causes. The research may face limitations in keeping up with these dynamic changes and adapting the AI system to evolving conditions. Continuous monitoring of the DN and regular updates to the AI model would be necessary to maintain its effectiveness. This could involve incorporating real-time data feeds, implementing feedback loops for model refinement, and considering methods for online learning to adapt to the evolving nature of the DN.

To address these limitations, researchers and practitioners should consider developing AI models with interpretability in mind, exploring techniques that provide transparency and insights into model decision-making processes. Additionally, they should establish mechanisms for continuous monitoring and updating of the AI system to ensure its effectiveness in capturing the evolving dynamics of the DN. This iterative and adaptive approach will contribute to better understanding and addressing the causes of abnormal line losses in DNs.

In addition, other critical limitations are related to external factors and computational requirements in the research on line losses of DNs.

1) Dependence on external factors: Line losses of DNs can be influenced by various external factors, including weather conditions, vegetation growth, equipment failures, and human interventions. If these factors are not adequately considered or accounted for in the research, the analysis and diagnosis of abnormal causes may be incomplete or inaccurate. Gathering relevant data and incorporating these external factors into the AI model is essential to ensure a comprehensive understanding of line losses and identify abnormal causes. This may require data integration from various sources and the development of sophisticated algorithms to account for the complex interactions between external factors and line losses.

2) Computational requirements and resource constraints: AI algorithms, especially deep learning models, can be computationally intensive and require substantial computational resources. Implementing and deploying these algorithms for real-time line loss analyses and diagnosis in DNs may be challenging due to computational limitations or resource constraints. The scalability and practical feasibility of the research may be limited by factors such as the availability of high-performance computing resources, data storage capabilities, and computational costs. Thus, it is essential to consider the computational requirements and resource constraints during the design and development of the AI system to ensure its practical applicability and efficiency.

Addressing these limitations requires a multidisciplinary approach integrating domain knowledge, data availability, and computational resources. Researchers and practitioners should collaborate closely with domain experts, utilities, and stakeholders to gather comprehensive data, incorporate external factors, and develop scalable and resource-efficient AI models. Additionally, there should be a continuous effort to evaluate and refine the models based on real-world feedback and operational constraints to ensure their practical applicability for line loss analyses and diagnosis in a DN.

Finally, the ethical considerations and the need for appropriate human involvement in decision-making processes when utilizing AI for analyzing and diagnosing abnormal causes in DNs.

While AI systems can provide valuable insights and assist in decision-making, it is crucial to maintain a balance between automated processes and human expertise. The research should address ethical considerations by recognizing the limitations of AI systems and the necessity of human oversight and intervention.

Here are a few key considerations:

1) Human-in-the-loop approach: The research should emphasize a human-in-the-loop approach, where AI is employed to augment human decision-making rather than replace it entirely. This involves designing the AI system to explain and justify its decisions, enabling humans to understand and validate the outputs. Human experts can bring domain knowledge, context-specific insights, and ethical judgment to the decision-making process, ensuring the final decisions align with legal, ethical, and regulatory requirements.

2) Transparent and interpretable AI: The research should strive to develop AI models that are transparent and interpretable. This allows human operators to understand the reasoning behind the AI system's recommendations or diagnoses. Interpretability helps build trust and confidence in the AI system and allows for better collaboration between humans and AI algorithms. Techniques such as explainable AI, model visualization, and feature importance analysis can contribute to the interpretability of AI models.

3) Accountability and responsibility: The research should consider mechanisms to ensure accountability and responsibility for the decisions made by the AI system. This includes documenting the decision-making processes, tracking the inputs and outputs of the AI system, and establishing protocols for reviewing and auditing its performance. In case of any errors or unintended consequences, there should be provisions for remedial actions and learning from these instances to improve the system's performance.

By addressing these ethical considerations and promoting appropriate human involvement, the research can develop AI systems that align with societal values, legal requirements, and ethical standards. This ensures that AI technology is employed to enhance decision-making and improve the management of abnormal causes in DNs while respecting human expertise and safeguarding against potential risks.

## CONCLUSION

To address the challenges that the current grid reality poses, the article introduces a novel approach for identifying anomalies of line loss data in the DN. The proposed technique leverages multidimensional characteristics and deviates from conventional theory-driven thinking in abnormal line loss identification. By combining the ISSA with the SVM, the abnormality identification model used for line losses in the DN achieves an impressive accuracy rate of up to 98%. Experimental analysis validates the effectiveness of the proposed method.

The key contributions of the article include:

1) Traceability model based on multidimensional features to detect the data abnormality of line losses daily: The article presents a comprehensive model that captures multidimensional features to trace and identify data abnormalities of line losses daily. This model enables a more holistic understanding of line loss patterns and aids in identifying the underlying causes of abnormalities.

2) Discovering the cause of line loss abnormality: By leveraging the multidimensional feature model, the proposed method facilitates discovering and understanding the factors contributing to line loss abnormalities. This knowledge is crucial for effective anomaly detection and targeted intervention strategies.

3) Optimizing the SVM parameters employing multiple strategy improvements of the SSA: The integration of the ISSA with the SVM allows for the optimization of the SVM parameters, such as $\sigma$ and c, to enhance the performance of the anomaly detection model. The proposed multiple strategy improvements of the ISSA algorithm enable efficient parameter search, leading to improved detection accuracy.

The article suggests that the proposed method can be applied to construct a system surveilling line losses of the DN. By leveraging this system, grid operators can enhance line loss management practices, ultimately leading to improved financial gains.

However, the article also acknowledges the limited availability of public data sets on line losses in DNs. As a result, further research is needed to determine whether the experimental sample data utilized in the research can be generalized to other voltage levels. Expanding the study to include additional voltage levels would enhance the technology and maximize its application performance based on a broader range of scenarios.

### Funding

This study was supported by the Science and Technology Project of the National Southern Power Grid of China (No. YNKJXM20220166). The funders had no role in study design, data collection and analysis, decision to publish, or preparation of the manuscript.

## Grant Disclosures

The following grant information was disclosed by the authors:

Science and Technology Project of the National Southern Power Grid of China: YNKJXM20220166.

## Competing Interests

The authors declare that they have no competing interests. All the authors are employed by Electric Power Research Institute, Yunnan Power Grid Co., Ltd.

## Author Contributions

- Yaohua Liao conceived and designed the experiments, performed the experiments, analyzed the data, performed the computation work, prepared figures and/or tables, authored or reviewed drafts of the article, and approved the final draft.
- Wang En conceived and designed the experiments, performed the experiments, analyzed the data, performed the computation work, prepared figures and/or tables, authored or reviewed drafts of the article, and approved the final draft.
- Bo Li conceived and designed the experiments, performed the computation work, prepared figures and/or tables, authored or reviewed drafts of the article, and approved the final draft.
- Mengmeng Zhu conceived and designed the experiments, performed the experiments, analyzed the data, performed the computation work, authored or reviewed drafts of the article, and approved the final draft.
- Bo Li conceived and designed the experiments, performed the experiments, analyzed the data, performed the computation work, authored or reviewed drafts of the article, and approved the final draft.
- Zhengxing Li performed the experiments, performed the computation work, prepared figures and/or tables, authored or reviewed drafts of the article, and approved the final draft.
- ZhiMing Gu performed the experiments, analyzed the data, performed the computation work, prepared figures and/or tables, authored or reviewed drafts of the article, and approved the final draft.

## Data Availability

Code and dataset which was used to implement the study is uploaded in supplementry section.

## Supplemental Information

Supplemental information for this article can be found online at http://dx.doi.org/10.7717/peerj-cs.1753#supplemental-information.

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
