# Peer review of "Research on line loss analysis and intelligent diagnosis of abnormal causes in distribution networks: artificial intelligence based method"

_PeerJ Computer Science, doi:10.7717/peerj-cs.1753_

## Round 0.1 · original submission · Major Revisions

Please revise the paper in light of the comments of experts and you are also required to improve the quality of English language and justify the contribution of the research. thank you.

**Language Note:** The Academic Editor has identified that the English language must be improved. PeerJ can provide language editing services - please contact us at [email protected] for pricing (be sure to provide your manuscript number and title). Alternatively, you should make your own arrangements to improve the language quality and provide details in your response letter. – PeerJ Staff

Reviewer 1 ·

Basic reporting

The idea presented in this paper titled "Research on key technology of distribution network line loss analysis and intelligent diagnosis of abnormal causes based on artificial intelligence" is good. However, the authors are suggested to address the following comments while revising the paper.

1. Why did the authors compute statistics such as mean, standard deviation, max, min, kurtosis, and skewness? Please discuss it.
2. Why did the authors implement the heuristic optimization method? Please discuss it.
3. Did the authors run other heuristic optimization methods to choose which one fits best? Please discuss it.
4. How did the authors decide to use ISSA over the others? Is there any empirical evidence? Please discuss it.

Experimental design

5. Which software is used to run the proposed method?
6. A section or a subsection should be allocated to the proposed method.
7. The steps of the proposed method should be restated.
8. What is the data set and how was it processed? Please discuss it.
9. How were the parameters initialized at the beginning? Please discuss it.
10. Did the authors run any normalization and preprocessing steps? Please discuss it.
11. Why did the authors set the upper iteration number to 100? Please discuss it.
12. What are the ratios of train and test datasets?

Validity of the findings

13. How could the authors explain that the operation time of SVM is the minimum? Please discuss it.

·

Basic reporting

The article consists of severe issues that require careful revisions. Each issue should be responded to clearly by the authors.
The issues are itemized as follows:
1. The title of the article should be rewritten, and the contribution of the article should be expressed in the title.
2. the abstract should be rewritten and reorganized. The logical order of the abstract should be enhanced.
3. Proofreading is a must.
4. More up-to-date references should be added and discussed.
5. The citation type does not follow a regular pattern. All citations should be checked and fixed.
6. The introduction section should be rewritten and reorganized. The research problem, research motivation, and contribution should be better expressed in separate paragraphs. Besides, the structure of the article should be expressed.
7. All titles of sections and subsections should be checked and fixed. Some of them are very long sentences.

Experimental design

8. All mathematical equations should be cited in the text. The terms used in those equations should be explained. The abbreviation, Eq.(.) should be used.
9. what is the statistical matrix? Please discuss it.
10. The proposed method and its steps should be presented in an algorithm.
11. the titles of figures and tables should be checked and fixed.
12. the numbering scheme of sections and subsections should be checked and fixed where necessary.
13. the conclusion section should be rewritten and reorganized. Besides, the future research agenda should be better stated. The advantages of the conducted research should be better underlined.

Validity of the findings

14. the discussion section actually contains both the results and the discussion. They need to be separated and better organized.
15. All AI methods used in the research are not optimized but SVM. How did the authors become confident that the optimized other AI methods used in the article would not generate better results than ISSA-SVM? Please discuss it.

---

## Round 0.2 · accepted · Accept

The reviewers have now reviewed your revised updated version of the article and based on their input, I'm pleased to inform you that your paper has been recommended for publication, congratulations!

Reviewer 1 ·

Basic reporting

Thank you for addressing the raised comments.

Experimental design

no comment

Validity of the findings

no comment

Additional comments

no comment

·

Basic reporting

no comment

Experimental design

no comment

Validity of the findings

no comment

Additional comments

no comment